# Characteristics of Gut Microbiota in Sows and Their Relationship with Apparent Nutrient Digestibility

**DOI:** 10.3390/ijms20040870

**Published:** 2019-02-18

**Authors:** Qing Niu, Pinghua Li, Shuaishuai Hao, Sung Woo Kim, Taoran Du, Jindi Hua, Ruihua Huang

**Affiliations:** 1Institute of Swine Science, Nanjing Agricultural University, Nanjing 210095, China; nqdwkx@163.com (Q.N.); lipinghua718@163.com (P.L.); 18114777806@163.com (S.H.); 2017205008@njau.edu.cn (T.D.); 2Huaian Academy of Nanjing Agricultural University, Huaian 223005, China; 3Department of Animal Science, North Carolina State University, Raleigh, North Carolina, NC 27695, USA; sungwoo_kim@ncsu.edu; 4Kunshan Second Animal Epidemic Prevention Station, Kunshan 215300, China; 5Suzhou Sutai Enterprise Co., Ltd., Suzhou 215000, China; 63huajd@163.com

**Keywords:** sows, gut microbiota, 16S rDNA, apparent nutrient digestibility, fiber

## Abstract

The gut microbiota plays important roles in animal health and nutrient digestibility. The characteristics of gut microbiota population in grower pigs and their correlation with apparent nutrient digestibility were assessed in previous study. Here we studied characteristics of intestinal microbiota of sows and analyzed their relationships with apparent nutrient (ether extract (EE), crude protein (CP), crude fiber (CF), neutral detergent fiber (NDF) and acid detergent fiber (ADF)) digestibility. *Firmicutes* and *Bacteroidetes* were the most dominant phyla, approximately 73% of the total sequences. *Treponema*, *Oscillibacter* and *Lactobacillus* were the most dominant generas, more than 49% of the total sequences. The microbiota of sows clustered separately from the microbiota of grower pigs at the age of D28 D60, D90 and D150. The abundance of *Clostridium* and *Turicibacter* was positively correlated with apparent EE digestibility. The abundance of *Anaerofustis* and *Robinsoniella* in sow fecal samples was positively correlated with apparent CF digestibility. The abundance of *Collinsella* and *Sutterella* was positively correlated with apparent NDF digestibility. The abundance of *Clostridium*, *Collinsella*, *Robinsoniella* and *Turicibacter* was positively correlated with apparent ADF digestibility. Sows have their unique gut microbial structure compared with grower pigs and some of them participate in the digestive process of different nutrients.

## 1. Introduction

A highly diversified community of microorganisms lives in the mammalian gastrointestinal tract with about 10^14^ microbes belong to approximately 500–1000 species [1,2,3]. Certain bacteria in the gut are known to be metabolically important for animal growth [4,5,6]. Gut microbiota participate in digestion, absorption, and metabolism of the host through the absorption of nutrients and the expulsion of metabolic wastes from the host [1,7,8]. Recent researches suggested that bacteria in the gut participated in lots of physiological actions such as obesity and metabolic disorders [9,10,11].

Pigs serve as important animal models for human diseases [12]. It might benefit for intestinal health and scientific diet of human beings to study the relationship between gut microbiota and nutrient digestibility in pigs. Pork is one of the main meat products consumed by human beings, accounting for about 60% of the total meat consumption in China. Therefore, for improving the efficiency of pork production it is beneficial to study the relationship between intestinal microbiota and digestibility of pigs. The gut microbiota are dynamic and vary with time, age and many other factors [2]. In our previous study, we have studied gut microbiota of fecal samples from the pigs in different growth phase at the age of D28, D60, D90 and D150, and reported the changes in the gut microbiota and apparent CF digestibility in pigs at different growth stages, as well as the correlation between the microbiota and apparent CF digestibility in pigs [12]. Sows have strong digestive capacity for various types of cellulose, proteins and fat macromolecules. Above all, dietary fiber has many benefits to the health and performance of sows. Digestion and absorption of fiber mainly occur in large intestine [13]. Adding a balanced amount of fiber in the sow diet can relieve stress and improve sow performance and nutrient balance [14,15]. The hypothesis was that gut of sows, compared with that of grower pigs, had a unique microbiota population related to apparent nutrient digestibility of sows. However, gut microbiota characteristics and their correlation with apparent nutrient digestibility of sows was still unclear.

The objective of this study was to assess the gut microbial structure and apparent nutrient digestibility in sows, as well as to explore the gut microbial communities related to nutrient digestibility.

## 2. Results

### 2.1. DNA Sequence Data and Bacterial Community Structure

A total of 1,677,029 paired-end 250 bp reads were acquired. The total read length was 1.75 gigabases (GB), and the average read length per sample was 0.19 GB. After initial quality control, 1,235,278 high quality sequences were obtained. On average, 137,253 sequences were obtained per sample. Based on 97% species similarity, 16,137 operational taxonomic units (OTUs) were obtained from samples of the sows.

Good’s coverage was 97.7%, suggesting that the present study captured the dominant phylotypes. The mean calculated values were 11,099 for Chao, 14,113 for Ace, 0.02 for Simpson and 5.58 for Shannon.

The results as shown in Figure 1 describe the distribution of DNA sequences in the level of phylum. A total of 25 phyla were shared by samples of the sows: *Acidobacteria*, *Actinobacteria*, *Armatimonadetes*, *Bacteroidetes*, *Chlamydiae*, *Chlorobi*, *Chloroflexi*, *Cyanobacteria, Deferribacteres*, *Deinococcus-Thermus*, *Elusimicrobia*, *Euryarchaeota*, *Fibrobacteres*, *Firmicutes*, *Fusobacteria*, *Gemmatimonadetes*, *Lentisphaerae*, *Nitrospira*, *Planctomycetes*, *Proteobacteria*, *Spirochaetes, Synergistetes, Tenericutes, TM7* and *Verrucomicrobia*. Among which, *Firmicutes* and *Bacteroidetes* were the largest shares (*p* < 0.01), comprising more than 73% of the total sequences, *Firmicutes* for more than 57% and *Bacteroidetes* for approximately 16%. The proportion of sequences that could not be assigned to a phylum using the Ribosomal Database Project (RDP) classifier was 22% in the fecal samples.

At the genus level, a total of 245 genera were identified from samples of the sows (Appendix A). The 10 most abundant genera, containing approximately 74% of the total sequences, were *Treponema*, *Oscillibacter*, *Lactobacillus*, *Clostridium sensu stricto*, *Bifidobacterium*, *Streptococcus*, *Fibrobacter*, *Clostridium XI*, *Roseburia* and *Ruminococcus*. One genus, *Bifidobacterium*, is a member of the phylum *Actinobacteria, Fibrobacter* belongs to the phylum *Fibrobacteres* and *Treponema* belongs to the phylum *Spirochaetes*. The other 7 genera belong to the phylum *Firmicutes*. Among them, *Treponema*, *Oscillibacter* and *Lactobacillus* were the most predominant genera, accounting for 17%, 15% and 12% of total sequences, respectively.

In addition to the phylum and genus, bacterial diversity and abundance were also analyzed at the level of class (Appendix A), order (Appendix A) and family (Appendix A). A total of 36 classes, 73 orders and 138 families were identified from nine samples, respectively. The most predominant class, order and family shared were *Clostridia, Clostridiales* and *Ruminococcaceae*, respectively.

The results as shown in Figure 2 described the genera distribution between D28, D60, D150 of growers and sows groups. A total of 9243 OTUs were shared by all the samples, and the unique OTUs number of sows group was much higher than other groups. From Figure 3, the microbiota of sows group clustered separately from the microbiota of other four groups (D28, D60, D90 and D150) along principal coordinate 1. The abundance of 5 genera, *Anaeroplasma*, *Bifidobacterium*, *Fibrobacter*, *Papillibacter* and *Treponema*, was significantly different between sows and the other 4 groups (Table 1).

### 2.2. Apparent Nutrient Digestibility in Sows and its Correlation with the Gut Microbiota

The ranges of apparent digestibility values (%) of EE, CP, CF, NDF and ADF were 55.34 ± 2.51, 88.44 ± 1.06, 61.61 ± 5.23, 65.83±4.02 and 54.47 ± 3.43, respectively (Table 2). Both of the apparent digestibility of EE and CP of sows were significantly different from that of each stage of grower pigs (D60, D90 and D150, *p* < 0.01). Both of the apparent digestibility of CF and ADF of sows were significantly different from that of two stages of grower pigs (D60 and D90, *p* < 0.01). The apparent digestibility of NDF of sows was significantly different from that of pigs in D60 group (*p* < 0.01).

At the genus level, based on Pearson’s correlation analysis, it showed that 2, 2, 2 and 4 genera were positively correlated with apparent EE, CF, NDF and ADF digestibility, respectively (Table 3). The bacterial abundance of *Clostridium* and *Turicibacter* were positively correlated with apparent EE digestibility. The bacterial abundance of *Anaerofustis* and *Robinsoniella* were positively correlated with apparent CF. The bacterial abundance of *Collinsella* and *Sutterella* were positively correlated with apparent NDF digestibility. The bacterial abundance of *Clostridium*, *Collinsella*, *Robinsoniella* and *Turicibacter* were positively correlated with apparent ADF digestibility. Moreover, compared with grower pigs, 2, 1, 2 and 3 unique genera were related with apparent digestibility of EE, CF, NDF and ADF in sows, respectively (Table 3).

## 3. Discussion

In our previous study, we got dynamic distribution of the gut microbiota and the relationship with apparent crude fiber digestibility and growth stages in pigs [12]. The goal of this study was to investigate characteristics gut microbiota and apparent nutrient digestibility in sows, as well as to elucidate the correlation between the microbiota and apparent nutrient digestibility in sows. So far, several studies have been published on the distribution of the gut microbiota across pig over life [2,3,4,16,17].

This study obtained a large number of gut microbiota data in sows and the read counts were greater than those in previous studies in pigs [4,16,17,18]. Moreover, according to Good’s coverage index (97%) of each sample, the modified sequences were comprehensive enough to cover most bacterial diversity. The number of total effective OTUs in this study was higher than that in the studies of Kim et al. [16], Ye and Zhang [19] and Whitehead and Cotta [20], indicating a much higher level of diversity than that reported in previous studies. The mean calculated values was 11,099 for shao, 14,113 for Ace, 0.02 for Simpson and 5.58 for Shannon. Chao and Ace indexes explored bacterial abundance. Simpson and Shannon indexes explored bacterial diversity. Zhao [21] studied the dynamic distribution of porcine microbiota across different ages and gastrointestinal tract segments. It showed that the rages of Chao, Ace, Simpson and Shannon index values 1 were 720–6674, 2222–7633, 0.01–0.54 and 1.86–5.67, respectively. Kim et al. [4] studied microbial shifts in the swine distal gut in response to the treatment with antimicrobial growth promoter, tylosin. They reported that the average Shannon–Weaver and Simpson index values per group were 4.87 and 0.96 for Farm 1, 5.28 and 0.97 for Farm 2. The range of these calculated values over the five sampling times was 4.39–5.50 for Shannon–Weaver and 0.95–0.98 for Simpson.

Using the RDP classifier, taxon-dependent analysis revealed that *Firmicutes* and *Bacteroidetes* were the most dominant phyla, representing approximately 73% of the total sequences. These results were similar to what others have observed in pig and human fecal samples [1,16,20]. Kim et al. [16] reported that the bacterial communities of all samples were primarily comprised of *Firmicutes* and *Bacteroidetes*, which accounted for more than 90% of the total sequences. Lamendella et al. [18] confirmed previous observations that most of the bacteria identified were in two phyla: *Firmicutes* and *Bacteroidetes*. Additionally, these two phyla comprised over 90% of the known phylogenetic categories, representing the dominant distal gut microbiota in the human intestinal tract [22]. From all the above results, the discrepancies between the present study and previous studies may have resulted from the use of pigs in different experiment design, diets, environmental conditions, breeds, ages, even the physiological stages, the length of the sampling time after feeding time, as well as the use of different sequencing platform and hypervariable region of the 16s rDNA (Appendix A) [2].

One of the objectives of the analysis of gut microbiota is to understand the metabolic contributions that the microbiota make to the physiology and metabolism of intestinal tract in the sows. The apparent digestibility of EE, CP, CF, NDF and ADF in pig fecal samples was analyzed in multiparity sows. The bacterial abundance of 2 genera (*Clostridium* and *Turicibacter*) was positively correlated with apparent EE digestibility. Woting et al. [23] showed that *Clostridium* ramosum promoted high-fat diet-induced obesity in gnotobiotic mouse models. Zhong et al. [24] reported that modulation of gut microbiota in rats fed high-fat diets by processing whole-grain barley to barley malt. They illuminated that *Turicibacter* was more abundant in the group assigned a high-fat diet (malt) and it correlated with butyric acid. Tilg and Kaser [25] described gut microbiome, obesity, and metabolic dysfunction. They showed that the gut microbiota interacts with host epithelial cells to indirectly control energy expenditure and storage, and gut microbiota enhances adiposity mainly by increased energy extraction from food and by regulating fat storage. Peter et al. [26] explored a core gut microbiome in obese and lean twins. Their results demonstrated that a diversity of organismal assemblages can nonetheless yield a core microbiome at a functional level, and that deviations from this core are associated with different physiologic states (obese versus lean). Jumpertz et al. [27] described the associations between nutrient absorption and the gut microbiota in humans, which indicated a possible role of the human gut microbiota in the regulation of nutrient harvest.

Ashida [8] and Tilg [25] showed that dietary fiber is mainly degraded by the gut microbiota. Gómez-Conde et al. [28] suggested a higher fermentation activity and possibly a change in the microbiota population in the animals fed increasing levels of soluble fiber. Chen et al. [29] indicated that long-term intake of fibrous diet could alter intestinal bacteria, mucosal digestive physiology and thus production parameters and/or health in fattening pigs. *Clostridium* is associated with dietary fiber metabolism [30] and *Turicibacter* correlated with butyric acid [23]. In this study, the bacterial abundance of *Clostridium* was positively correlated with apparent CF and ADF digestibility and *Turicibacter* was positively correlated with apparent ADF digestibility.

Compared with our previous study [12], the results showed in Table 2 illustrated that the apparent nutrient digestibility of sows were much higher than those in grower pigs. The PCoA analysis showed that microbiota of sows clustered separately from the microbiota of grower pigs along principal coordinate 1. Compared with groups of grower pigs, the bacterial abundance of five genera, *Anaeroplasma*, *Bifidobacterium*, *Fibrobacter*, *Papillibacter* and *Treponema* were much higher in sow group. *Bifidobacteria* belong to probiotics, which can improve intestinal health [31]. *Fibrobacteres* is a small bacterial phylum which includes many of the major rumen bacteria, allowing for the degradation of plant-based cellulose in ruminant animals [32]. Sow has a different physiological stages compared with grower pigs. Most fiber fermentation occurs in the large intestine of pig [33]. Sow has well-developed intestinal microbiota, higher microbial activity and stronger fiber fermentation [34,35,36]. The final end-products of microbial fermentation of carbohydrates are short-chain fatty acids (SCFAs). SCFAs have the functions of supplying energy and maintain health of the epithelia [37,38]. These results indicated that sow had a healthier intestinal environment [39] and greater intestinal capacity [36] than those of grower pigs. Moreover, comparing the related genera with nutrients, it was quite different between sow and grower pigs groups, 2, 1, 2 and 3 unique genera were related with apparent digestibility of EE, CF, NDF and ADF in sow fecal samples, respectively. These results illuminated that sow has its specific gut microbiota structure and functions.

This manuscript supplemented the data of gut microbiota in sows and their relationship with nutrients digestibility. These information can provide reference for optimizing pig feeds or intestinal health. Just studying gut microbiota is a first level. Gut microbiota functionality is a much more relevant level, also related to gut development and gut health. For further research, we will focus on discovering those gut microbiota of pigs able to degrade fiber and studying their energy supply mechanism to the host.

## 4. Materials and Methods

### 4.1. Animals and Sample Collection

Fecal and diet samples from 9 healthy Sutai multiparous sows (3rd–5th parity, empty stage (pregnancy and lactation stages had a significant effect on the gut microbiota), no disease or diarrhea occurred at least one week before sampling) were randomly collected under the same husbandry condition at Suzhou Sutai Enterprise Co., Ltd., Suzhou, China. All Sutai pigs [40,41] were selected by using a unified breeding standard. For Sutai pig, the body weight of gilts at the age of 6 months is 70–85 kg. Animals were grouped based on a randomized block experimental design and fed a corn-soybean non-antibiotic diet. The diet ingredients are provided in the Appendix A (Appendix A).

Sows were individually raised in different pens in the same house and samples were randomly selected from different pens. Fecal samples were collected by sterile 2 mL centrifuge tubes without any treatment, and these samples were used for 16S rDNA gene sequencing analysis. Fecal samples were collected by plastic bags (approximately 200 g of each fecal sample) were fixed on site by mixing with 15 mL 10% sulfuric acid, and these samples were used for analyzing apparent nutrient digestibility. All the collected samples were kept cool in an ice box for transportation and then stored at −20 °C in the laboratory before DNA extraction and apparent nutrient digestibility assessment [12].

All procedures involving animals were carried out in accordance with the Guide for the Care and Use of Laboratory Animals prepared by the Institutional Animal Care and Use Committee of Nanjing Agricultural University, Nanjing, China. All experimental protocols were approved by the Animal Care and Use Committee of Nanjing Agricultural University [42,43].

### 4.2. 16S rDNA Sequencing: DNA Extraction, PCR Amplification, Amplicon Sequence, Sequence Data Processing, Taxonomy Classification and Statistical Analysis

Microbial genomic DNA was extracted from fecal samples using the TIANGEN DNA stool mini kit (TIANGEN, cat#DP328). The V4 hypervariable regions of 16S rDNA were amplified by PCR using the barcoded fusion primers. The barcoded fusion forward primer was 520F 5-AYTGGGYDTAAAGNG-3, and the reverse primer was 802R 5-TACNVGGGTATCTAATCC-3. The PCR condition was as follows: initial denaturation at 94 °C for 4 min; 94 °C denaturation for 30 sec, 50 °C annealing for 45 sec, and 72 °C extension for 30 sec, repeated for 25 cycles; final extension at 72 °C for 5 min. PCR production from each sample was used to construct sequencing library by using TruSeq™ DNA Sample Prep kit-SetA (Illumina, San Diego, CA, USA). Sequences with an average phred score lower than 30, ambiguous bases, homopolymer runs exceeding 6 bp, primer mismatches, or sequence lengths shorter than 100 bp were removed. Only sequences with an overlap longer than 10 bp and without any mismatch were assembled according to their overlap sequence. Reads that could not be assembled were discarded.

### 4.3. Experimental Feeds and Chemical Analysis

Diet samples were collected in plastic bags and stored at −20 °C. Pigs did not receive antibiotics in the feed or for any therapeutic purposes.

Fecal samples from sows were dried at 65 °C to a constant weight. Acid insoluble ash (AIA) was used as an indigestible marker to assess the digestibility of the dietary components (AOAC 942.05) [44]. The EE content was measured by using the soxhlet extraction method (AOAC 920.85), which was performed with a soxhlet apparatus. The CP content was measured based on the Kjeldahl method (AOAC 984.13) using a Kjeltec 8400 analyzer unit (Foss, zhongguancun south street, Beijing, China). Analysis of CF, NDF and ADF digestibility was carried out by using the ANKOM A200 filter bag technique (AOAC 962.09) [45].

The contents of different nutrient compositions were calculated as follows:
(1)$$\mathrm{CF}\left(\%\right)=\frac{100\times (M2-(M1\times C1))}{Ma}$$
(2)$$\mathrm{NDF}\left(\%\right)\frac{100\times (M4-(M3\times C2))}{Mb}$$
where: *M*a = Sample weight; *M*1 = Bag tare weight; *M*2 = Weight of organic matter (the loss of weight upon ignition of the bag and fiber); *C*1 = Ash-corrected blank bag factor (a running average of the loss of weight upon ignition of the blank bag/original blank bag)
(3)$$\mathrm{NDF}\left(\%\right)=\frac{100\times (M4-(M3\times C2))}{Mb}$$
(4)$$\mathrm{ADF}\left(\%\right)=\frac{100\times (M5-(M3\times C3))}{Mb}$$
where: *M*b = Sample weight; *M*3 = Bag tare weight; *M*4 = Weight of organic matter after extraction by neutral detergent; *M*5 = Weight of organic matter after extraction by acid detergent; *C*2 = Ash-corrected blank bag factor (a running average of the loss of weight after extraction of the blank bag/original blank bag); *C*3 = Ash-corrected blank bag factor (a running average of the loss of weight after extraction of the blank bag/original blank bag)
(5)$$\mathrm{EE}\left(\%\right)=\frac{100\times (M6-M7)}{Mc}$$
where: *M*c = Dry sample weight; *M*6 = Filter paper weight before extraction; *M*7 = Filter paper weight after extraction. The digestibility of each sample diet was calculated by the indicator technique [46] according to the equation:
(6)$${\mathrm{CAD}}_{D}\left(\%\right)=100\times \left(1-(DC_{F}\times AIA_{D}))/((DC_{D}\times AIA_{F})\right)$$
where CAD_D_ is the coefficient of the apparent digestibility of dietary components in the assay diet; *DC_F_* is the dietary component concentration in feces; *AIA_D_* is the *AIA* concentration in the assay diet; *DC_D_* is the dietary component concentration in the assay diet; and *AIA_F_* is the *AIA* concentration in feces.

## 5. Data Analysis

Data of apparent nutrient digestibility and gut microbiota population in pigs at the age of D28, D60, D90 and D150 in our previous study [12] were used for comparing with data of sows. All fecal samples of D28, D60, D90, D150 and sows were collected on one day and 16S rDNA sequencing analysis was completed at the same time. The sex of 3 piglets at the age of 28 was male, and all the other samples were from female pigs. All effective bacterial sequences were compared to the RDP database (Release 11.1 http://rdp.cme.msu.edu/) (access on 16th June 2003) using the best hit classification option to classify the abundance count of each taxon [47,48]. The sequence length was archived by QIIME [49]. The relationship between the selected taxonomy group (abundant phyla, genera, classes, orders or families), the observed OTUs or the bacterial community index (Chao1) and the apparent nutrient digestibility was calculated using SPSS 13.0 software [50]. Independent sample test was used for data different analysis in Table 1 and Table 2. Venn diagram (http://bioinfogp.cnb.csic.es/tools/venny/) (access on 9th March 2015), as software of Venn diagram can only make pairwise comparisons between 4 groups and gut microbial diversity in D60 was similar with that in D90, the data of D90 was not used to make a Venn diagram analysis. PCoA and comparative analysis were used to explore the gut microbiota population in different groups. Weighted clustering was performed using PCoA of UniFrac distance matrices. Pearson’s correlations were used to assess the associations between bacterial abundance and apparent nutrient digestibility [12].

## 6. Conclusions

Sows have their unique gut microbial structure compared with grower pigs (pigs at the age of D28, D60, D90 and D150) and some of them participate in the digestive process of different nutrients.

## Figures and Tables

**Figure 1 ijms-20-00870-f001:**
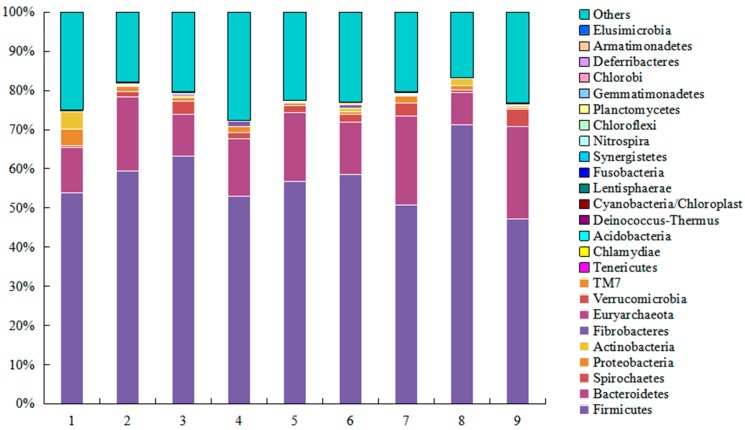
Phylum distribution of gut microbiota. Distribution of the phylum as a percentage of the total number of identified 16S rDNA sequences from fecal samples of the nine sows.

**Figure 2 ijms-20-00870-f002:**
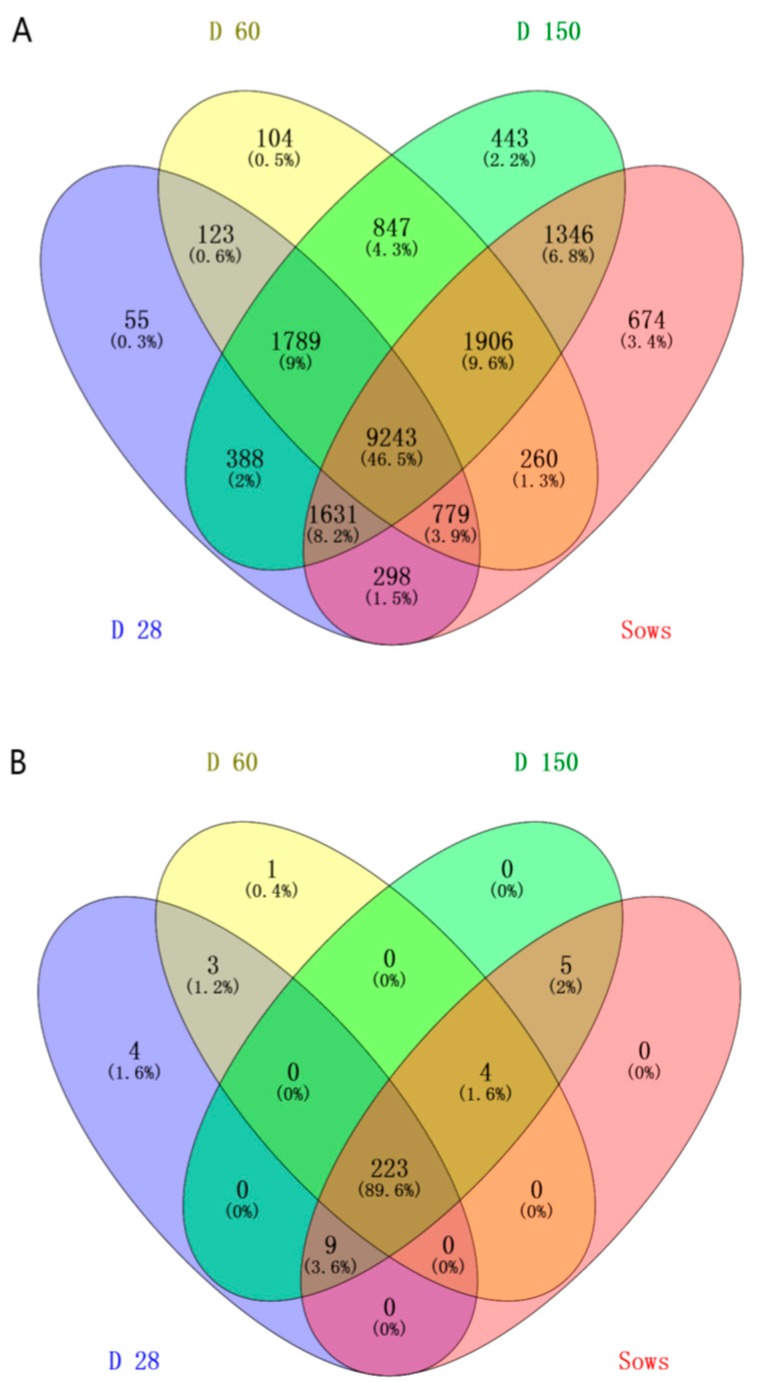
A Venn diagram of the OTUs. The Venn diagrams mainly showed OTUs (**A**) and genera (**B**) of sows group separately compared with groups of grower pigs at the age of D28, D60 and D150 and depicted the unique OTUs and genera of 4 groups.

**Figure 3 ijms-20-00870-f003:**
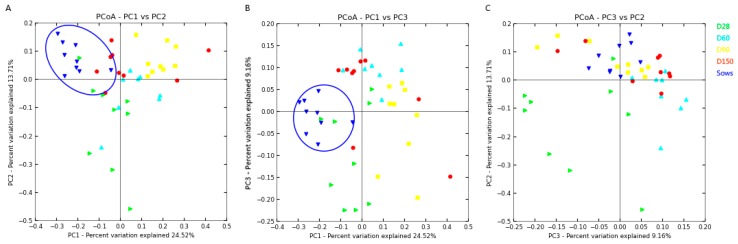
The comparison of bacterial community components between different aged groups. Principal co-ordinates analysis (PCoA) of weighted UniFrac distances for the fecal microbiota at different phases. In the weighted UniFrac analysis of the fecal samples, the first principal coordinate, explained 24.52% of sample variation, separated sows group from others (D28, D60, D90 and D150 groups).

**Table 1 ijms-20-00870-t001:** Different genera of between sows and the other four groups.

	28D	60D	90D	150D	Sows
Treponema	241.0 ± 63.8 ^B^	202.5 ± 60.4	803.9 ± 194.0	1670.8 ± 923.7	4212.4 ± 601.6 ^A^
Bifidobacterium	188.5 ± 42.6 ^b^	65.9 ± 26.8 ^b^	135.5 ± 76.6 ^b^	97.2 ± 27.6 ^b^	1080.0 ±600.7 ^a^
Fibrobacter	19.0 ± 2.2 ^B^	22.3 ± 4.7 ^B^	103.4 ± 42.9 ^B^	53.7 ± 10.6 ^B^	590.0 ± 241.5 ^A^
Anaeroplasma	2.8 ± 0.4 ^B^	7.8 ± 4.9 ^B^	7.4 ± 3.4 ^B^	35.6 ± 6.9 ^B^	141.1 ± 38.9 ^A^
Papillibacter	0.4 ± 0.2 ^B^	0.5 ± 0.3 ^B^	0.3 ± 0.2 ^B^	0.3 ± 0.2 ^B^	20.2 ± 6.6 ^A^

^a^ and ^b^ represent the mean difference is significant at a level of 0.05; ^A^ and ^B^ represent the mean difference is significant at a level of 0.01.

**Table 2 ijms-20-00870-t002:** Comparison of apparent nutrient digestibility between sows and grower pigs.

	Sows	D60	D90	D150
**EE**	55.34 ± 2.50 ^A^	26.83 ± 3.77 ^B^	20.83 ± 3.78 ^B^	26.97 ± 2.73 ^B^
**CP**	88.45 ± 1.05 ^A^	73.66 ± 1.14 ^B^	77.18 ± 0.90 ^B^	77.39 ± 0.68 ^B^
**CF**	61.61 ± 5.23 ^Aa^	37.05 ± 2.49 ^Bab^	49.11 ± 3.14 ^AB^^b^	63.10 ± 1.78 ^ABa^^b^
**NDF**	65.83 ± 4.02 ^a^	56.74 ± 2.29 ^b^	69.35 ± 1.48 ^ab^	62.43 ± 3.53 ^ab^
**ADF**	54.47 ± 3.43 ^Aa^	28.46 ± 2.71 ^Bab^	30.53 ± 2.48^Bab^	44.55 ± 2.70 ^ABb^

Apparent EE, CP, CF, NDF and ADF digestibility of sows were compared with those of grower pigs at the age of D60, D90 and D150, respectively. ^a^ and ^b^ represent the mean difference is significant at a level of 0.05; ^A^ and ^B^ represent the mean difference is significant at a level of 0.01.

**Table 3 ijms-20-00870-t003:** Comparison of genera correlated with apparent nutrient digestibility in fecal samples of sows and grower pigs.

	Genera (Pearson’s Positive Correlation)	
	Sows	Grower Pigs
**Apparent EE digestibility**	*Clostridium* (0.780 *) *Turicibacter* (0.723 *)	
**Apparent CF digestibility**	*Anaerofustis* (0.735 *) *Robinsoniella* (0.799 **)	*Anaeroplasma* (0.595 **)
		*Campylobacter* (0.511 **)
		*Clostridium* (0.453 **)
		*Enterococcus* (0.415 *)
		*Janibacter* (0.397 *)
		*Methanobrevibacter* (0.568 **)
		*Nitrosospira* (0.433 *)
		*Propionibacterium* (0.500 *)
		*Pseudobutyrivibrio* (0.505 *)
		*Robinsoniella* (0.606 **)
		*Staphylococcus* (0.402 *)
		*Treponema* (0.542 **)
		*Turicibacter* (0.418 *)
**Apparent NDF digestibility**	*Collinsella* (0.716 *) *Sutterella* (0.744 *)	*Methanobrevibacter* (0.449 *)
		*Parasporobacterium* (0.469 *)
		*Sporobacter* (0.435 *)
		*Treponema* (0.553 **)
**Apparent ADF digestibility**	*Clostridium* (0.761 *) *Collinsella* (0.757 *) *Robinsoniella* (0.768 *) *Turicibacter* (0.781 *)	*Anaeroplasma* (0.481 *)
		*Campylobacter* (0.437 *)
		*Caulobacter* (0.443 *)
		*Cloacibacillus* (0.546 **)
		*Enterococcus* (0.434 *)
		*Lactobacillus* (0.496 *)
		*Methanobrevibacter* (0.561 **)
		*Nitrosospira* (0.488 *)
		*Propionibacterium* (0.461 *)
		*Pseudomonas* (0.455 *)
		*Robinsoniella* (0.651 **)
		*Staphylococcus* (0.476 *)
		*Treponema* (0.542 **)

* The correlation is significant at a level of 0.05; ** the correlation is significant at a level of 0.01.

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
