# Peer review of "Characteristics of Gut Microbiota in Sows and Their Relationship with Apparent Nutrient Digestibility"

_ijms, 2019, doi:10.3390/ijms20040870_

Round 1
Reviewer 1 Report
Overall, the manuscript presents new data on the relation between gut microbiota of growing pigs and sows and nutrient digestion, which mainly is of interest to microbiologists, scientists in veterinary medicine and experts in animal nutrition.
This reviewer still struggles with the relevance of this work for this specific journal and its audience. The authors are asked why they have submitted this work to IJMS and why a more veterinary journal wasn't considered? The work described by the authors is not really demonstrating new molecular techniques, insight or novelties, which might be slightly out of scope for the IJMS' audience.
The quality of the manuscript can be overall improved. Below some specific comments and suggestions.
Specific comments:
-The manuscript can still benefit from stating even more clearly what this study is aimed to bring or show? Is it still for the creating of more microbiota data and filling / completing databases or could it have any implications or possible benefits, e.g. towards optimising animal feeds or animal health?
-English needs to have a better check. Some examples:
-phylas (plural of phylum is phyla)
-line 43: ‘digestive capacity’ might be better than ‘digestion power’
-line 44: ‘dietary fiber’ instead of ‘diet fiber’
-line 45-46: ‘…were mainly occurred…’?
-line 113: ‘…significantly different’ instead of ‘…significant different’
-When reading the abstract only, it’s not clear what EE-digestibility means, neither CF, NDF nor ADF digestibilities. Please spell out these abbreviations in the abstract.
-What is the total number of samples taken? E.g. line 66 ‘…shared by all samples‘, line 80 ‘…identified from all samples’, line 93 ‘…shared by all the samples’.
-Line 90: what is the origin of the 9 samples and the characteristics of the sows (age, sex, body weight, time points, etc)? Suggestion to provide this information in a small table.
-Fig 1: the legend doesn’t explain the different bars at the X-axis (1-9) in the figure. It seems these are the number of samples. What is indicated by ‘…in individual groups’?
-Page3: explanation about figure 2 is very short and unclear. What is compared, what is ‘sows’ vs. the different phases?
-Fig 3: an explained variation of ~10% (PC1) is not high. Its relevance should be more clearly addressed in the Discussion section.
-Line 92: it would be better to specify more clearly D28, D60 and D150 related to growing pigs vs. sows. When are they adult / mature?
-Discussion section: what is the relevance of these data? As above, are the authors able to elaborate a bit on potential benefits, optimisation or new approaches for animal health / welfare / nutrition? Can the authors give some more suggestions for further research? Why is there no further discussion about the role of SCFAs or other metabolites in this context? Just studying gut microbiota is a first level, gut microbiota functionality is a much more relevant level, also related to gut development and gut health.
-Table 2: not clear how the comparisons are made on the statistics with marks like A, B, a and b. Which groups are compared and to what comparisons do the marks refer to?
-Line 195-197: would not refer to Wikipedia, better to use scientific papers when referring to Fibrobacteres relevance.
-Line 198: not clear on what ‘…a healthier intestinal environment…’ is based on. It needs a scientific substantiation and reference.
-From line 168 – line 187: a lot of referencing to other studies, most of them in other animals (rodents) models and humans, but not clear why or to what context? Also the authors should be very careful to compare microbial communities between different species, they differ largely.
-Lines 205-206: please specify how ‘healthy’ and ‘without disease’ is defined. Authors should also declare whether antibiotics were used (currently or in the past) or not at all since these have a large influence of microbiota composition.
-Line 296-298: Not sure if the journal requires a separate Conclusion-section. Otherwise, this conclusion could be merged with the Discussion section.
-Under data analyses (e.g. line 280-295) the type of statistical tests aren’t clearly specified, e.g. for statistically significant differences in tables 1 and 2.
Author Response
Comments and Suggestions for Authors
Overall, the manuscript presents new data on the relation between gut microbiota of growing pigs and sows and nutrient digestion, which mainly is of interest to microbiologists, scientists in veterinary medicine and experts in animal nutrition.
1. This reviewer still struggles with the relevance of this work for this specific journal and its audience. The authors are asked why they have submitted this work to IJMS and why a more veterinary journal wasn't considered? The work described by the authors is not really demonstrating new molecular techniques, insight or novelties, which might be slightly out of scope for the IJMS' audience.
Re: Thank you very much for your comments. Our work is about gut microbial structure of pigs, apparent nutrient digestibility and their relationship with gut microbiota which is related to several research topics included in this special issue, Such as
l Role of microbiota-food interaction in the maintenance of human health.
l Modulation of intestinal microbiota and beneficial effects of functional foods or specific dietary ingredients.
l Experimental studies on the mechanisms involved in the microbiota-food-health interaction.
Pigs serve as important animal models for human diseases. It might benefit for intestinal health and scientific diet of human beings to study the relationship between gut microbiota and nutrient digestibility in pigs. Pork is one of the main meat products consumed by human beings, accounting for about 60% of the total meat consumption in China. Therefore, it is benefit for improving the efficiency of pork production to study the relationship between intestinal microbiota and digestibility of pigs. Line 38-42
The quality of the manuscript can be overall improved. Below some specific comments and suggestions.
Specific comments:
2. -The manuscript can still benefit from stating even more clearly what this study is aimed to bring or show? Is it still for the creating of more microbiota data and filling / completing databases or could it have any implications or possible benefits, e.g. towards optimising animal feeds or animal health?
Re: Very great comments. We completely agree with you. This manuscript supplemented the data of gut microbiota of sows and their relationship with nutrients digestibility. These information can provide reference for optimizing pig feeds or intestinal health.
Pigs serve as important animal models for human diseases. It might benefit for intestinal health and scientific diet of human beings to study the relationship between gut microbiota and nutrient digestibility in pigs. Pork is one of the main meat products consumed by human beings, accounting for about 60% of the total meat consumption in China. Therefore, it is benefit for improving the efficiency of pork production to study the relationship between intestinal microbiota and digestibility of pigs. Line 38-42
3. -English needs to have a better check. Some examples:
-phylas (plural of phylum is phyla)
Re: This is a basic grammar mistake we’ve made. Thank you so much for mentioned it.It has been revised. Line 19
-line 43: ‘digestive capacity’ might be better than ‘digestion power’
Re: Thank you so much. It has been revised. Line 47
-line 44: ‘dietary fiber’ instead of ‘diet fiber’
Re: Thank you so much. It has been revised. Line 48
-line 45-46: ‘…were mainly occurred…’?
Re: Thank you so much. ‘…mainly occur…’has been instead of ‘…were mainly occurred…’. Line 49
-line 113: ‘…significantly different’ instead of ‘…significant different’
Re: Thank you so much. It has been revised. Line 97, 115 and 116
4. -When reading the abstract only, it’s not clear what EE-digestibility means, neither CF, NDF nor ADF digestibilities. Please spell out these abbreviations in the abstract.
Re: It’s really our carelessness and they have been added. Line 17-18. As the abstract should be a total of about 200 words maximum. ‘Digestibility of sows was thought strong for various types of cellulose, proteins and fat macromolecules.’ was deleted.
5. -What is the total number of samples taken? E.g. line 66 ‘…shared by all samples‘, line 80 ‘…identified from all samples’, line 93 ‘…shared by all the samples’.
Re: Thank you for your comments. We have added the total number of samples taken in the revision. Fecal samples from 9 sows were sampled in the present study according to the material methods introduced in Line 221. Now ‘all samples’ in line 64, 69 and 80 were revised to ‘samples of the sow ’.
‘all samples’ in line 234 was revised to ‘all the collected samples’.
‘…shared by all the samples’ in line 93 were samples from ‘D28, D60, D150 and sow groups’.
6. -Line 90: what is the origin of the 9 samples and the characteristics of the sows (age, sex, body weight, time points, etc)? Suggestion to provide this information in a small table.
Re: Thank you for your comments. Fecal and diet samples from 9 healthy Sutai multiparous sows (3rd-5th parity, empty stage (Pregnancy and lactation stages had a significant effect on the gut microbiota), no disease or diarrhea occurred at least one week before sampling) were randomly collected under a same husbandry condition at Suzhou Sutai Enterprise Co., Ltd, Suzhou, China. Line 221-224. We had not measured the body weight of the sows. The sows had reached their sexual maturity and body maturity and the stage of these sows is at empty stage of the 3rd-5th parity.
7. -Fig 1: the legend doesn’t explain the different bars at the X-axis (1-9) in the figure. It seems these are the number of samples. What is indicated by ‘…in individual groups’?
Re: Thank you for your carefulness. It has been revised to ‘…from fecal samples of the 9 sows’. Line 79
8. -Page3: explanation about figure 2 is very short and unclear. What is compared, what is ‘sows’ vs. the different phases?
Re: How important of this statement, thank you very much. The Venn diagram showed pairwise comparisons of the four groups. It mainly showed the results of sows separately compared with grower pigs at the age of D28, D60 and D150. ‘The Venn diagrams mainly showed OTUs (A) and genera (B) of sows group separately compared with groups of grower pigs at the age of D28, D60 and D150 and depicted the unique OTUs and genera of 4 groups.’ has been added. Line 100-102
‘‘sows’ vs. the different phases’ means the data from samples of sows and grower pigs at the age of D28, D60, D150 were compared.
9. -Fig 3: an explained variation of ~10% (PC1) is not high. Its relevance should be more clearly addressed in the Discussion section.
Re: Thank you so much for the important reminds. We have corrected the algorithm of UniFrac distance. Unweighted UniFrac distance algorithm didn't count relative abundance of sample sequences in different environments, while weighted UniFrac algorithm weighted the richness of sequences when calculating the length of branches, so unweighted UniFrac can detect the existence of changes among samples
Weighted clustering was performed using PCoA of UniFrac distance matrices. Line 306
The explained variation of PC1 is 24.52%. Line 107-111
10. -Line 92: it would be better to specify more clearly D28, D60 and D150 related to growing pigs vs. sows. When are they adult / mature?
Re: Thank you for your comments. ‘For Sutai pig, the body weight of gilts at the age of 6 months is 70-85kg.’ has been added. Line 225-226
11. -Discussion section: what is the relevance of these data? As above, are the authors able to elaborate a bit on potential benefits, optimisation or new approaches for animal health / welfare / nutrition? Can the authors give some more suggestions for further research? Why is there no further discussion about the role of SCFAs or other metabolites in this context? Just studying gut microbiota is a first level, gut microbiota functionality is a much more relevant level, also related to gut development and gut health.
Re: Thank you so much for the important reminds, discussion has been added as suggested, as follows.
The objective of this study was to assess the gut microbial structure and apparent nutrient digestibility in sows, as well as to explore the gut microbial communities related to nutrient digestibility. Line 55-57
Sow has a different physiological stages compared with grower pigs. Most fiber fermentation occurs in the large intestine of pig. Sow has well-developed intestinal microbiota, higher microbial activity and stronger fiber fermentation. The final end-products of microbial fermentation of carbohydrates are short-chain fatty acids (SCFAs). SCFAs have the functions of supplying energy and maintain health of the epithelia. Line 203-207
Just studying gut microbiota is a first level, gut microbiota functionality is a much more relevant level, also related to gut development and gut health. For further research, we will focus on discovering those gut microbiota of pigs able to degrade fiber and stuyding their energy supply mechanism to the host. Line 215-218
Reference:
Jha, R.; Berrocoso, J. F. D., Dietary fiber and protein fermentation in the intestine of swine and their interactive effects on gut health and on the environment: A review. Anim Feed Sci Tech 2016, 212, 18-26.
Renteria-Flores, J. A.; Johnston, L. J.; Shurson, G. C.; Moser, R. L.; Webel, S. K., Effect of soluble and insoluble dietary fiber on embryo survival and sow performance. J anim sci 2008, 86, (10), 2576-84.
Fernández, J. A.; Jørgensen, H.; Just, A., Comparative digestibility experiments with growing pigs and adult sows. Anim Prod 2010, 43, (01), 127-132.
Goff, G. L.; Milgen, J. v.; Noblet, J., Influence of dietary fibre on digestive utilization and rate of passage in growing pigs, finishing pigs and adult sows. Anim Sci 2016, 74, (03), 503-515.
Musso, G.; Gambino, R.; Cassader, M., Interactions between gut microbiota and host metabolism predisposing to obesity and diabetes. Annu rev med 2011, 62, 361-80.
Suzuki, T.; Yoshida, S.; Hara, H., Physiological concentrations of short-chain fatty acids immediately suppress colonic epithelial permeability. Brit j nutr 2008, 100, (2), 297-305.
12. -Table 2: not clear how the comparisons are made on the statistics with marks like A, B, a and b. Which groups are compared and to what comparisons do the marks refer to?
Re: Thank you, ‘Apparent EE, CP, CF, NDF and ADF digestibility of sows were compared with those of grower pigs at the age of D60, D90 and D150, respectively.’ It has been added in line 128-129
Independent sample test was used for data difference analysis in Table 1 and Table 2. Line 306-307
13. -Line 195-197: would not refer to Wikipedia, better to use scientific papers when referring to Fibrobacteres relevance.
Re: Thank you so much for the important reminds. New reference has been added. Line 203
Reference:
Ransom-Jones, E.; Jones, D. L.; McCarthy, A. J.; McDonald, J. E., The Fibrobacteres: an important phylum of cellulose-degrading bacteria. Microb ecol 2012, 63, (2), 267-81.
14. -Line 198: not clear on what ‘…a healthier intestinal environment…’ is based on. It needs a scientific substantiation and reference.
Re: Thank you so much for the important reminds. In the healthy intestine, these bacterial communities reside in defined anatomical locations and exist in a symbiotic relationship with their hosts, promoting normal physiologic processes and limiting colonization with potentially pathogenic microbes.
New reference has been added. Line 208-209
Reference:
Goff, G. L.; Milgen, J. v.; Noblet, J., Influence of dietary fibre on digestive utilization and rate of passage in growing pigs, finishing pigs and adult sows. Anim Sci 2016, 74, (03), 503-515.
Bach Knudsen, K. E.; Hedemann, M. S.; Lærke, H. N., The role of carbohydrates in intestinal health of pigs. Anim Feed Sci Tech 2012, 173, (1-2), 41-53.
15. -From line 168 – line 187: a lot of referencing to other studies, most of them in other animals (rodents) models and humans, but not clear why or to what context? Also the authors should be very careful to compare microbial communities between different species, they differ largely.
Re: Very great comments. we completely agree with you. However, studies on the function of pig specific microbial species have not been fully reported. Here we want to express the potential functions of interested gut microbiota through literature. References to human studies were referred in present study. Pigs serve as an important model organism because of their similarity to humans at the anatomical, physiological and genetic level, making them very useful for studying a variety of human diseases.
Reference:
Fang XD et al., The sequence and analysis of a Chinese pig genome. GigaScience 2012, 1:16
16. -Lines 205-206: please specify how ‘healthy’ and ‘without disease’ is defined. Authors should also declare whether antibiotics were used (currently or in the past) or not at all since these have a large influence of microbiota composition.
Re: Very great comments. In this manuscript ‘healthy’ and ‘without disease’ are the same meaning. It means the sows did not have diarrhea or other diseases.
In order to avoid the impact of antibiotics on gut microbiota, we did not add antibiotics in thes diet of sows. At the same time, Sutai pig is a Chinese new lean type pig breed, and we does not add antibiotics in the diet of Sutai sows in the actual production process. ‘Sows fed a corn-soybean non-antibiotic diet’ was shown in line 226-227.
17. -Line 296-298: Not sure if the journal requires a separate Conclusion-section. Otherwise, this conclusion could be merged with the Discussion section.
Re: The requirement of the journal is as following.
Conclusions: This section is not mandatory, but can be added to the manuscript if the discussion is unusually long or complex.
In this case, independent conclusions were retained.
18. -Under data analyses (e.g. line 280-295) the type of statistical tests aren’t clearly specified, e.g. for statistically significant differences in tables 1 and 2.
Re: Thank you very much for your reminding. Independent sample test was used for data difference analysis in Table 1 and Table 2. Line 306-307
Reviewer 2 Report
The present work is dealing with the correlation among sows-gut microbiota and nutrient digestibility. The topic is interesting, and work is well designed and performed. In my opinion, the Discussion section should be improved. It is more a repetition of the results than a critical discussion of them. The reasons for the correlations between microbiota and digestion should be more in deep discussed. Why do microbiota and digestive capacities differ between sows and growing pigs? What is the practical meaning of this work?
Minor corrections
P1 L16: Do you mean “hard digestion” or “efficient digestion capability”?
P1 L24; L26; L27: Please, explain the meaning of the acronyms to allow the reader to understand the meaning of the abstract without reading the whole manuscript.
P1 L41: Please, explain the meaning of the acronyms the first time you use them.
P2 L48: What do you mean for saw performance? Please, clarify.
P2 L69: Please define TM7.
P7 L164-165: Please, avoid repetition (lines 152-153).
P7 L198: What do you mean for healthier environment? The only presence of Bifidobacterium and Fibrobacter is enough to state that?
Author Response
Comments and Suggestions for Authors
1. The present work is dealing with the correlation among sows-gut microbiota and nutrient digestibility. The topic is interesting, and work is well designed and performed. In my opinion, the Discussion section should be improved. It is more a repetition of the results than a critical discussion of them. The reasons for the correlations between microbiota and digestion should be more in deep discussed. Why do microbiota and digestive capacities differ between sows and growing pigs? What is the practical meaning of this work?
Re: Very great comments. Line 141
Sow has a different physiological stages compared with grower pigs. Most fiber fermentation occurs in the large intestine of pig. Sow has well-developed intestinal microbiota, higher microbial activity and stronger fiber fermentation. Line 203-206
This manuscript supplemented the data of gut microbiota of sows and their relationship with nutrients digestibility. These information can provide reference for optimizing pig feeds or intestinal health.
Pigs serve as important animal models for human diseases. It might benefit for intestinal health and scientific diet of human beings to study the relationship between gut microbiota and nutrient digestibility in pigs. Pork is one of the main meat products consumed by human beings, accounting for about 60% of the total meat consumption in China. Therefore, it is benefit for improving the efficiency of pork production to study the relationship between intestinal microbiota and digestibility of pigs. Line 38-42
Minor corrections
2. P1 L16: Do you mean “hard digestion” or “efficient digestion capability”?
Re: Thank you for your attention. Sows have an ‘efficient digestion capability’ about nutrient digestibility.
By the way, since the abstract is limited in no more than 200 words in maximum. ‘Digestibility of sows was thought strong for various types of cellulose, proteins and fat macromolecules.’ was deleted.
3. P1 L24; L26; L27: Please, explain the meaning of the acronyms to allow the reader to understand the meaning of the abstract without reading the whole manuscript.
Re: It’s really our carelessness and they have been added. Line 17-18
4. P1 L41: Please, explain the meaning of the acronyms the first time you use them.
Re: Thank you very much and it’s really our carelessness. Crude fiber (CF) have been added in the abstract. Line 18
5. P2 L48: What do you mean for saw performance? Please, clarify.
Re: Sorry for our carelessness, it should be ‘sow performance’, means performance related with litter size and total litter weight at birth and weaning. Line 50
6. P2 L69: Please define TM7.
Re: Seven-span transmembrane (TM7) is the result of a database annotation and it was reported that TM7 receptors represent the largest gene family in animal genomes. TM7 receptors respond to a diverse array of sensory and chemical stimuli, such as light, taste, odor, pheromones, calcium ions, neurotransmitters, hormones and chemokines.
Reference: Kwakkenbos, M. J.; Kop, E. N.; Stacey, M.; Matmati, M.; Gordon, S.; Lin, H. H.; Hamann, J., The EGF-TM7 family: a postgenomic view. Immunogenetics 2004, 55, (10), 655-66.
7. P7 L164-165: Please, avoid repetition (lines 152-153).
Re: Thank you very much for your carefulness. It has been revised. Line 165-169
8. P7 L198: What do you mean for healthier environment? The only presence of Bifidobacterium and Fibrobacter is enough to state that?
Re: Thank you so much for the important reminds. Bifidobacterium is one of the most important probiotics, it has biologic and healthy functions. Fibrobacter degrades fiber in pig gut to provide energy for the host. In theses cases, results in the present study showed sows had a healthier intestinal environment than growing pigs.
Best regards,
Ruihua Huang
Round 2
Reviewer 1 Report
no further comments